# NEURAL ODEs FOR IMAGE SEGMENTATION WITH LEVEL SETS

## ABSTRACT

We propose a novel approach for image segmentation that combines Neural Ordinary Differential Equations (NODEs) and the Level Set method. Our approach parametrizes the evolution of an initial contour with a NODE that implicitly learns from data a forcing function describing the evolution. In cases where an initial contour is not available or to alleviate the need for careful choice or design of contour embedding functions, we propose using NODEs to directly evolve the embedding of an input image into a pixel-wise dense semantic label. We evaluate our methods on kidney segmentation (KiTS19) and on salient object detection (PASCAL-S, ECSSD and HKU-IS). In addition to improving initial contours provided by deep learning models while using a fraction of their number of parameters, our approach achieves $F_\beta$ scores that are higher than several state-of-the-art deep learning algorithms.

## 1 INTRODUCTION

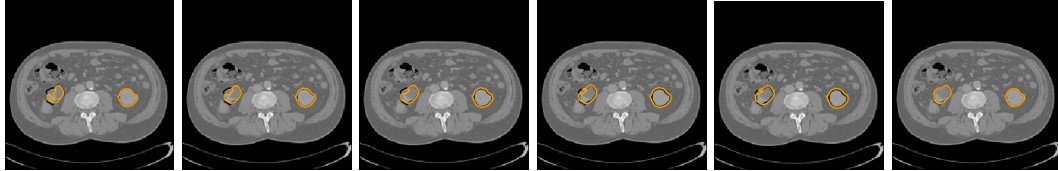

Figure 1: Transversal slices of CT scans. Leftmost image: initial contour provided by a UNet model. Other images: intermediate steps of the evolution of the initial contour with our Neural ODE model.

Image segmentation is the task of delineating pixels belonging to semantic labels. The ability to automatically segment objects is important because accurate labeling is expensive and hard (Vittayakorn & Hays, 2011; Zhang et al., 2018). Automatic image segmentation can have large impact in many domains, e.g. obstacle avoidance in autonomous driving and treatment planning in medical imaging.

Accurate classification of pixels in close proximity to inter-class boundaries remains a challenging task in image segmentation. Object boundaries can have high curvature contours or weak pixel intensity that complicate separating the object from surrounding ones. In deep CNNs (Simonyan & Zisserman, 2014; Zeiler & Fergus, 2014; Szegedy et al., 2015; He et al., 2016; Chen et al., 2017), the object-of-interest and surrounding competing objects can provide equal context to a receptive field of a boundary pixel, which can make accurate classification difficult. Humans also find it difficult to accurately label pixels near object boundaries.

Level Set methods (Zhao et al., 1996; Brox et al., 2006) and Active Shapes (Paragios & Deriche, 2000; Chan & Vese, 2001) have been proposed to incorporate shape and image priors to mitigate boundary ambiguities (Tsai et al., 2003; Rousson & Paragios, 2002). The Level Set method for image segmentation evolves an initial contour of an object-of-interest along the normal direction with a forcing function. A contour is represented by an embedding function, typically a signed distance function, and its evolution amounts to solving a differential equation (Osher & Sethian, 1988).

In this work, we extend the formulation of the level set method. Inspired by the recent progress in Neural Ordinary Diferential Equations (NODEs) (Chen et al., 2018; Dupont et al., 2019; Gholami et al., 2019), we propose to use NODEs to solve the level set formulation of the contour evolution, thus learning the forcing function in an end-to-end data driven manner. Unlike earlier attempts in combining the level set method with CNNs, we benefit from NODEs parametrization of the derivative of the contour because it allows us to incorporate external constraints that guide the contour evolution, e.g. by adding a regularization penalty to the curvature of the front or exploiting images at the evolving front by extracting appearance constraints in a non-supervised way.

Finally, similar to experiments in (Chen et al., 2018), to alleviate the need for careful choice or design of contour embedding functions, we propose a NODE-based method that evolves an image embedding into a dense per-pixel semantic label space.

To the best of our knowledge, this work is the first to apply Neural ODEs to real world problems. We validate our methods on two 2D segmentation tasks: kidney segmentation in transversal slices of CT scans and salient object segmentation. Given an initial estimate of kidney via existing algorithms, our method effectively evolves the initial estimates and achieves improved kidney segmentation, as we show in Figure 1. On real life salient objects, in addition to contour evolution, we use our method to directly evolve the embedding of an input image into a pixel-wise dense semantic label.

Following (Hu et al., 2017), we compare against the results in (Wang et al., 2017; Li et al., 2016; Li & Yu, 2015; Zhao et al., 2015; Lee et al., 2016; Wang et al., 2015; Hu et al., 2017) and achieve $\omega$-$F_\beta$ scores, PASCAL-S 0.668 and ECSSD 0.768, that are higher than several state-of-the art algorithms. Our results suggest the potential of utilizing NODEs for solving the contour evolution of level set methods or the direct evolution of image embeddings into segmentation maps. We hope our findings will inspire future research in using NODEs for semantic segmentation tasks. We foresee that our method would allow for intervention on intermediate states of the solution of the ODE, allowing for injection of shape priors or other regularizing constraints.

In summary, our contributions are:

- We propose to use NODEs to solve the level set formulation of the contour evolution.
- We propose using NODEs to learn the forcing function in an end-to-end data driven way.
- We show NODEs can also evolve image embeddings directly into dense per-pixel semantic label spaces, which may alleviate the need for careful choice or design of contour embedding functions.

## 2 METHODS

Suppose $\mathbf{I}$ is a 2D image, $\mathbf{S}$ is the contour of an object we want to segment, and $\phi$ is a contour embedding function, defined as a distance map, such that $\mathbf{S} = \{(x, y)|\phi(x, y) = 0\}$. We assume an initial but rough contour of the object is given by a human operator or by an existing algorithm. A level set segmentation (Osher & Sethian, 1988) solves a differential equation to evolve a contour along its normal direction with a speed function $F$ as:

$$\frac{d\phi}{dt} = |\nabla\phi|F \text{ for } t \in [0, 1], \tag{1}$$

where the initial value $\phi^0(x, y)$ is defined as a signed Euclidean distance from $(x, y)$ to the closest point on the initial contour $\mathbf{S}^0$. The speed function $F$ is often modelled to be a function of the target image $\mathbf{I}$, the shape statistics of the object contour (derived from training shapes), or a regularizing curvature term ($\nabla\frac{\nabla\phi}{\|\nabla\phi\|}$).

In Neural ODEs, we parametrize the derivative of the hidden state $h$ using a neural network $f_\theta$ parametrized by $\theta$:

$$\frac{dh}{dt} = f_\theta(h, t). \tag{2}$$

The relationship between Eq. 1 and Eq. 2 is immediate. In the next section, we propose two approaches that adapt NODEs to the level set method for image segmentation.

## 2.1 Contour Evolution with NODEs

We propose to solve a more general form of Eq. 1 to evolve an initial contour estimate $\hat{\phi}$ for image segmentation. We define the state of the NODE to be $\hat{\phi}$ augmented with the input image's embedding, $h$. We then advance the augmented state, $\gamma = (\hat{\phi}, h)$, using NODEs, which can be interpreted as estimating the speed function $F$ described in Eq. 1. Mathematically,

$$
\begin{aligned}
\gamma &= (\hat{\phi}, h), \\
\frac{d\gamma}{dt} &= f_\theta(\gamma, t) \ \text{ for } t \in [0, 1], \\
\gamma^{(0)} &= (\hat{\phi}^{(0)}, h^{(0)}), \\
\tilde{\phi} &= \hat{\phi}^{(1)} + \psi(\gamma^{(1)}),
\end{aligned}
\tag{3}
$$

where $t$ is the time step in the evolution, $\gamma$ is the augmented state of the NODE, $f$ is a neural network parametrized by $\theta$, $\hat{\phi}^{(0)}$ is the initial value of the distance map, $h^{(0)}$ is the initial value of the image embedding, $\psi$ is a learned function and $\tilde{\phi}$ is the dense per-pixel distance map prediction. Figure 2a schematically illustrates our initial contour evolution approach. Throughout this paper, we will refer to this method as **Contour Evolution**.

## 2.2 Image Evolution with NODEs

In our first approach, we obtain a final optimal contour by evolving an initial estimate. In our second approach, inspired by Chen et al. (2018), we evolve an image embedding $h$ and project it into a dense per-pixel distance map $\tilde{\phi}$, whose zero level set defines the final segmentation map, $\mathbf{S}^{(t)} = \{(x, y) | \phi^{(t)}(x, y) = 0\}$ . Mathematically,

$$
\begin{aligned}
\frac{dh}{dt} &= f_\theta(h, t) \text{ for } t \in [0, 1], \\
h^{(0)} &= \lambda(I), \\
\tilde{\phi} &= \psi(h^{(1)}),
\end{aligned}
\tag{4}
$$

where $t$ is the time step in the evolution, $f$ is a neural network parametrized by $\theta$, $I$ is an image, $\lambda$ is a learned image embedding function and $\psi$ is a learned function that maps an image embedding to a distance map. Figure 2b schematically illustrates our direct image evolution approach. Throughout this paper, we will refer to this method as **Image Evolution**.

## 3 Implementation

In the following subsections, we describe our design choices in loss function and their regularization terms, architectures, strategies for emphasizing the evolution of the contour on a region of interest. We also detail our model initialization strategies to prevent drifting from the sub-optimal initial value, and choices of error tolerances and activation normalization.

### 3.1 Loss function and regularization terms

We optimize the parameters of our NODE models, described in Figures 2a and 2b, to minimize the empirical risk computed as the Mean Squared Error (MSE) between the target ($\phi$) and predicted ($\tilde{\phi}$) distance maps. We remind the reader that although our techniques can access intermediate NODE states, which could allow injection of priors or other constraints, we do not explore this in our current experiments, and relay it to future work.

### 3.2 Narrow band and Re-initialization

In the level set formulation, all levels that describe the propagating contour are tracked. Adalsteinsson & Sethian (1995) proposed limiting the evolution to the subset of levels within a narrow band of the zero level contour.

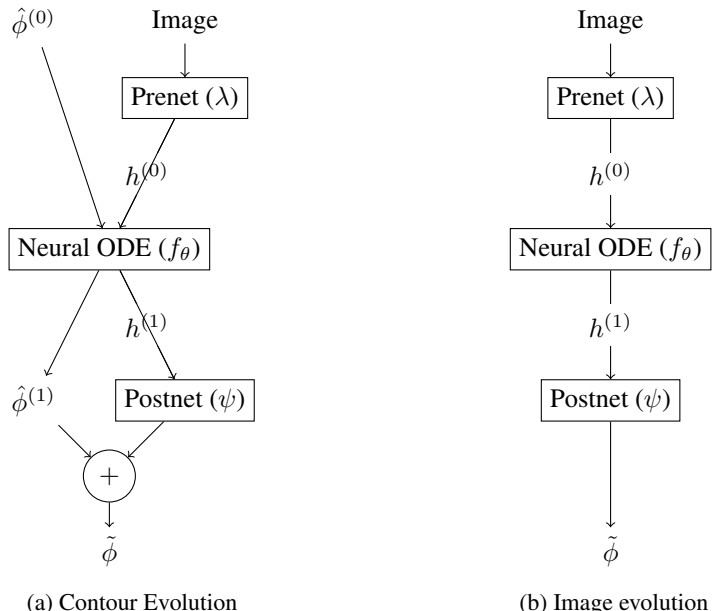

(a) Contour Evolution  (b) Image evolution

Figure 2: Diagrams for the contour and image evolution methods described in sections 2.1 and 2.2. Superscripts 0 and 1 represent initial value and numerical solution of the NODE respectively.

In our approach, we obtain the equivalent of a narrow band by applying a hyperbolic tangent non-linearity on the evolved distance map. It effectively attenuates the contribution of levels in the optimization process. This transformation is especially valuable in refinement setups because it weights the gradients of the loss according to the proximity to the contour[1].

Re-initialization of $\phi$ is another common practice in classical level set methods. It ensures the states in the trajectory of the numerical solution remain valid distance maps. (Sussman et al., 1994; Hartmann et al., 2010) propose to first extract a zero level set of an evolving state, and re-calculate a distance map of that contour. In our experiments, we found that our optimization is not sensitive to non valid distance maps, and we did not find it necessary to reinitialize $\phi$.

### 3.3 PARAMETER INITIALIZATION AND LEARNING RATE RAMPUP

In tasks where the initial value is already close to the desired solution, not initializing the model parameters to represent the identity function and not using learning rate ramp up can slow down the optimization process as the model predictions can immediately drift away from the initial value.

In addition to using learning rate rampup, we prevent this issue by setting the weights and biases on the **last layer** of the NODE and Postnet layers to zero. This approach has been successfully used in normalizing flow models (Kingma & Dhariwal, 2018; Prenger et al., 2019).

### 3.4 ADAPTIVE SOLVERS AND ERROR TOLERANCES

In ordinary differential equations, adaptive step solvers vary the step size according to the error estimate of the current step and the error tolerance. If the error estimate is larger than the threshold, the step will be redone with a smaller size until the error is smaller than the error tolerance.

The error tolerance $e^i_{tol}$ given the current state $i$ is the sum of the absolute error tolerance $a_{tol}$ and the infinity norm of the current state $h$ weighted by the relative error tolerance $r_{tol}$:

$$e^i_{tol} = a_{tol} + r_{tol} * \left\| h^i \right\|_\infty . \tag{5}$$

---

[1]For the hyperbolic tangent, the gradients decrease as it moves away from zero, which represents the contour at the zero level set.

Given that we do not know in advance the infinity norm of $h^i$, which in our case contains the image embedding as described in Equations 3 and 4, we set the contribution of the relative error tolerance term to zero and adjust the absolute error tolerance.

### 3.5 ACTIVATION NORMALIZATION

When the batch size is too small for using batch size dependent normalization schemes like Batch-Norm (Ioffe & Szegedy, 2015), researchers rely on dataparallel multi-processor training setups with BatchNorm statistics reduced over all processes, for example SyncBatchNorm in the APEX library (Sarofeen et al., 2019).

When training in multi-processor environments with data parallelism and NODEs with adaptive step solvers, the number of NODE function evaluations on each processor can differ. Consequently, the number of BatchNorm calls inside each NODE layer will be dependent on the number of function evaluations, thus making reduction over processes complex.

In our setup, we circumvent this issue by using GroupNorm (Wu & He, 2018) in layers where no convolution groups are used and LayerNorm (Ba et al., 2016) when convolution groups are used.

## 4 EXPERIMENTS

### 4.1 DATASETS AND TASKS

The KiTS19 Challenge Data (Heller et al., 2019) consists of CT scans from 210 patients with tumour and kidney annotations. The MSRA10K dataset (Cheng et al., 2014) consists of 10000 images with pixel-level saliency labeling from the MSRA dataset. The PASCAL-S dataset (Li et al., 2014) consists of free-viewing fixations on a subset of 850 images from PASCAL VOC, the ECSSD dataset (Yan et al., 2013) consists of 1000 semantically meaningful but structurally complex images and the HKU-IS (Li & Yu, 2015) consists of 4447 including multiple disconnected salient objects.

For the kidney segmentation task, we prune images that do not contain a kidney and resize the data to 200 by 200, without any loss of generality given that the original images are interpolated with the same affine transformation for each patient. We divide the dataset between 7108 images from 168 patients for training and 1786 images from another 42 patients for validation.

For the salient object detection task, we train on the MSRA10K dataset and use 512 by 512 image crops, padding where necessary and masking the loss accordingly. We use data augmentation procedures such as scale, horizontal flip and change in brightness. We use all images for training and compute validations scores on the other salient object detection datasets.

### 4.2 EVALUATION METRICS

In our experiments we use three metrics: Intersection Over Union (IOU), adaptive $F_\beta$ ($\alpha$-$F_\beta$) and weighted $F_\beta$ ($\omega$-$F_\beta$) described in (Margolin et al., 2014).

For computing IOUs, we rely on the definition from the PASCAL-VOC challenge (Everingham et al., 2015) and compute it as $T_P/(T_P + F_P + F_N)$, where $T_P$, $F_P$, and $F_N$ represent true positive, false positive and false negative pixels determined **over the whole validation set**.

The $\alpha$-$F_\beta$ metric is computed as the weighted $F_1$ score, $F_1 * (1 + \beta^2)/\beta^2$, where we set $\beta^2 = 0.3$ (Hu et al., 2017) and compute $F_1$ over the entire validation set . For computing the weighted $F_\beta$, we use the MatLab code provided by (Margolin et al., 2014), compute scores per image and average over all images. We understand these are the mechanism used to compute the scores reported in Hu et al. (2017).

### 4.3 TRAINING SETUP

All our models are trained in PyTorch (Paszke et al., 2017). We use the Adam optimizer (Kingma & Ba, 2014) with default params and learning rates between 1e-3 and 1e-4. We anneal the learning rate once the loss curves start to plateau.

We use the Runge-Kutta (RK-45) adaptive solver and the adjoint sensitivity method provided in (Chen, 2019). We set the relative error tolerance to zero and explore absolute error tolerances between 1e-3 and 1e-5.

The model architectures we evaluate include the UNet (Ronneberger et al., 2015), NODEs parametrized by UNets (NodeUnet), and an architecture inspired by DeepLabV3 (Chen et al., 2017), in which we stack NODEs (NodeStack) with Squeeze and Excitation modules (Hu et al., 2018) followed by an Atrous Spatial Pyramid Pooling Layer (Chen et al., 2017).

The kidney segmentation experiments were conducted on a single NVIDIA DGX-1 with 8 GPUs. The salient object detection experiments on UNet and NodeUNet were conducted on a single NVIDIA DGX-1 with 8 GPUs, and the experiments on NodeStack were conducted on 4 NVIDIA DGX-1 nodes with 32 GPUs total. We used the largest possible batch size given memory constraints.

The code for replicating our experiments and pre-trained weights will be made available on github.

## 4.4 RESULTS

In this section, we provide comparative results between our methods (*contour evolution* and *image evolution*) and other methods over multiple datasets. In our experiments, our contour evolution method focuses on using a NODE to refine suboptimal contours obtained from a regression model trained to predict distance maps from an image; our image evolution method focuses on using a NODE to learn to evolve an image embedding into a distance map. We first provide results on kidney segmentation and then provide results on salient object detection.

### 4.4.1 KIDNEY SEGMENTATION

In this task we compare three setups: the first trains a UNet *regression* model that maps an image to a distance map; the second uses our *contour evolution* method to refine the prediction of the aforementioned UNet model with a NODE parametrized by a UNet (NodeUNet); the third uses our *image evolution* method to train a NodeUnet that evolves an image into a distance map.

We chose the UNet model checkpoint used in the contour evolution experiment by selecting the checkpoint with the lowest validation loss on the first 8 samples of the validation set right before the UNet starts overfitting the training data and the validation loss starts going up.

We use the same training and validation setup for all models and provide results over the validation set below on Table 1. The UNet models provide the worse IOU scores. Our image evolution method produces goods results, showing evidence that it is possible to use NODEs to evolve an image into a distance map.

Lastly, our contour evolution method is able to improve the suboptimal initial contour provided by the UNet model and represents our best results in this experiment. These promising results show evidence that our method could generally be used to improve suboptimal models that underfit or overfit the training data. This is specially valuable for domains with scarcity of data. We provide results of the NodeUNet model trained with the contour evolution method in Figure 3.

| Model | Method | Parameters | IOU |
|---|---|---|---|
| UNet | Regression | 5M | 0.8762 |
| UNet | Regression | 15M . | 0.8494 |
| NodeUNet | Image Evolution | 5M | *0.8832* |
| NodeUNet | Contour Evolution | 5M | **0.8866** |

Table 1: Validation IOU scores from three methods using similar model architectures with similar and different number of parameters. Image Evolution represents evolution from an image to a distance map and Contour Evolution represents refinement of an initial contour.

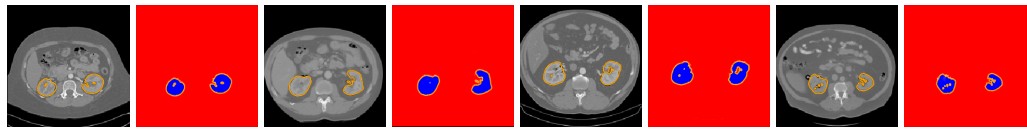

Figure 3: Kidney segmentation results. Black and orange contours are ground truth and model prediction, respectively. Blue, white and orange images are narrow band distance map prediction.

### 4.4.2 SALIENT OBJECT DETECTION

In this task we replicate the training setup described in Hu et al. (2017): we train our models on the MSRA10K dataset and compute their validation $F_\beta$ scores on PASCAL-S, ECSSD and HKU-IS.

We first evaluate the effect of different methods and model architectures on $F_\beta$ scores. We compare results from a *regression* model trained using the UNet architecture, a *contour evolution* model using the NodeUNet architecture and an *image evolution* model using the NodeStack architecture. We choose the UNet model for contour evolution by repeating the procedure described in 4.4.1.

We provide results over the validation set below on Table 2, wherein the UNet model provides the baseline $F_\beta$ scores. In all experiments, our refinement method provides 5% relative improvement over our UNet baseline, which has 3x more paremeters than our NodeUNet (5M vs 15M). We foresee that this trend will continue and models with more parameters will yield better refinement results.

| Dataset | Metrics | UNet 15M Regression | NodeUNet 5M Contour Evolution | NodeStack 41M Image Evolution |
|---------|---------|---------|---------|---------|
| PASCAL-S | $\alpha - F_\beta$ | 0.745 | *0.762* | **0.780** |
| PASCAL-S | $\omega - F_\beta$ | 0.696 | **0.704** | *0.698* |
| ECSSD | $\alpha - F_\beta$ | 0.736 | *0.771* | **0.848** |
| ECSSD | $\omega - F_\beta$ | 0.719 | *0.745* | **0.768** |
| HKU-IS | $\omega - F_\beta$ | 0.672 | *0.701* | **0.734** |

Table 2: Scores for different methods and models with 15, 5 and 41 million parameters. All scores are computed on binary maps produced by thresholding the ground truth saliency maps at $0$.

Finally, we compare our best model against a contrast based model DRFI (Wang et al., 2017) and recent deep learning based models such as MTDS (Li et al., 2016), MDF (Li & Yu, 2015), MCDL (Zhao et al., 2015), ELD (Lee et al., 2016), LEGS (Wang et al., 2015). We also compare against DLS (Hu et al., 2017), a deep learning model based on level sets.

Whenever possible, we use the original code provided by the authors for computing scores or collect the scores from their publications. We reproduce the setup in Hu et al. (2017) by computing our PASCAL-S scores on binary maps produced by thresholding the ground truth saliency maps at $0.5$. Figure 4 below illustrate our model performance on PASCAL-S, ECCSD and HKU-IS.

Table 3 below shows that the NodeStack model (**Ours**) achieves the best results on all but one metric.

| Dataset | Metrics | DRFI | MCDL | LEGS | MTDS | MDF | ELD | DLS | Ours |
|---------|---------|------|------|------|------|-----|-----|-----|------|
| PASCAL-S | $\alpha - F_\beta$ | $\sim 0.6$ | $\sim 0.7$ | $\sim 0.7$ | $\sim 0.7$ | $\sim 0.7$ | $\sim 0.7$ | $\sim 0.7$ | **0.740** |
| PASCAL-S | $\omega - F_\beta$ | 0.514 | 0.573 | 0.596 | 0.537 | 0.582 | *0.658* | 0.651 | **0.668** |
| ECSSD | $\alpha - F_\beta$ | $\sim 0.7$ | $\sim 0.8$ | $\sim 0.8$ | $\sim 0.8$ | $\sim 0.8$ | $\sim 0.8$ | $\sim 0.8$ | **0.848** |
| ECSSD | $\omega - F_\beta$ | 0.0 | 0.679 | 0.682 | 0.663 | 0.692 | 0.756 | *0.766* | **0.768** |
| HKU-IS | $\omega - F_\beta$ | 0.514 | 0.634 | 0.607 | 0.711 | 0.567 | 0.718 | **0.748** | *0.734* |

Table 3: $F_\beta$ Scores. PASCAL-S scores are computed on binary maps produced by thresholding the ground truth saliency maps at $0.5$.

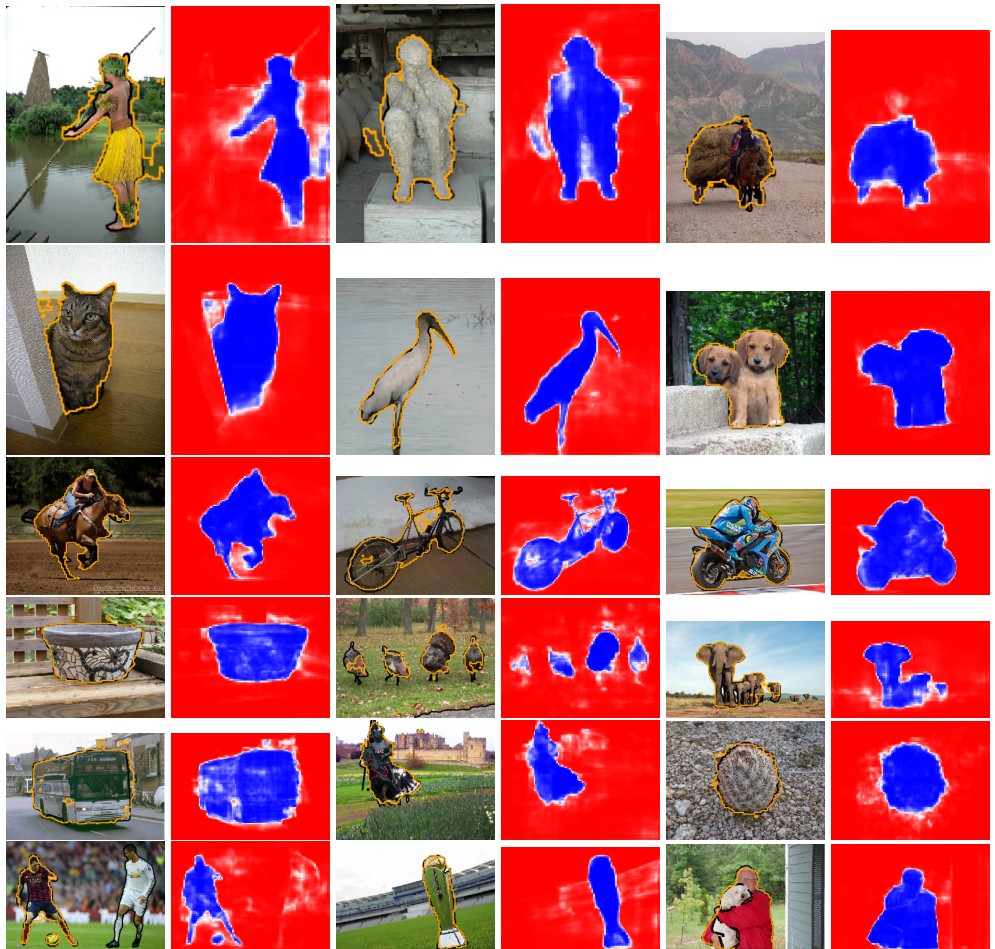

Figure 4: Salient object results. Black and orange contours are ground truth and model prediction, respectively. Blue, white and orange images are narrow band distance map prediction.

## 5   DISCUSSION

In this paper, we extend the level set segmentation method to use NODEs to solve the contour evolution problem. We learn a forcing function in an end-to-end data driven manner. We demonstrate that our techniques can effectively evolve rough estimates of contours into final segmentation of objects. Our techniques can also evolve input image's embedding into a pixel-wise dense semantic label.

Experimental results on several benchmark datasets suggest using NODEs for image segmentation task is viable. Compared to state-of-the-art methods, our proposed techniques also produce favourable segmentation results.

Although we benefit from NODEs' parametrization of the derivative of the contour, in this paper we do not explore the incorporation of external constraints to guide the contour evolution and that is an area for future exploration. We also foresee that our method can generalize to 3D images.

Finally, in some cases during our hyperparameter search we found that training the same model architecture with different learning rates and error tolerances yielded similar losses but largely different number of NODE function evaluations, prohibitively increasing wall clock time. This hyperparameter search can be replaced with automated approaches such as the Gated Info CNF described in Nguyen et al. (2019) , where the error tolerances are estimated by the model.

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
