# OpenReview forum: "Neural ODEs for Image Segmentation with Level Sets"
_ICLR.cc/2020/Conference — Reject_

### Official Review · AnonReviewer3 · 2019-10-16
**Official Blind Review #3**

**Rating:** 3

**Review:**

This paper proposes to utilize Neural ODEs (NODEs) and the Level Set Method (LSM) for the task of image segmentation.  The argument is that the NODE can be used to learn the force function in an LSM and solve the contour evolution process. The authors propose two architectures and demonstrate promising performance on a few image segmentation benchmarks.

While I like the attempt to combine deep learning with traditional CV algorithms, I do not think this submission is suitable for acceptance. My major critique is that both the two proposed models (Figure 2) are essentially two fully differentially NODE based architecture trained in a purely supervised way (by minimizing MSE). This makes the flavor of the proposed methods drastically different from what an LSM does, whereas the latter heavily rely on rich priors embedded in the design of the force function. To be more specific, the image evolution method (Figure 2(b)) essentially has nothing to do LSM. The contour evolution method bares more similarity to LSM but only in the sense that it learns to iteratively refine a contour estimation with an NODE. In the experimental evaluations, it seems that the image evolution model works favorably compared to the contour evolution method. This suggests that the main benefit of the proposed method comes from applying an NODE based architecture to a supervised learning task, rather than the inductive prior brought by LSM.

Given this, I think a much more proper way of presenting this work should be from the view of applying an NODE to the supervised image segmentation task. This reduces the novelty but increases clarity, and may still make the work a useful empirical reference for these benchmarks.

Some more detailed comments:
1. Equation 3&4 are not super easy to follow. For example, \gamma^{(1)} is not defined or explained in main text.
2. A related work section will be useful, especially for readers who are not familiar with LSM and NODE related literature.
3. Experiments with more careful control needed. For example, from Table 1, it seems that the baseline Unet 15M performs inferior to the 5M model (explanation needed too), while Table 2 only compares against the 15M model. This makes it difficult to interpret the results.

**Experience Assessment:**

I do not know much about this area.

**Review Assessment: Checking Correctness Of Derivations And Theory:**

I assessed the sensibility of the derivations and theory.

**Review Assessment: Checking Correctness Of Experiments:**

I assessed the sensibility of the experiments.

**Review Assessment: Thoroughness In Paper Reading:**

I read the paper at least twice and used my best judgement in assessing the paper.

---

### Official Review · AnonReviewer2 · 2019-10-23
**Official Blind Review #2**

**Rating:** 1

**Review:**

This paper proposes to apply the Neural ODE framework (Chen et al 2018) for image segmentation. The method relies on contour delineation through Level Sets. Since contour estimation requires to solve an ODE, this naturally allows to apply the work presented in (Chen et al 2018). The method is here applied in two segmentation tasks: kidney segmentation and salient object detection.

The concept underlying the paper is interesting, and leverages on very recent advances in the field. The idea of learning the dynamics required to evolve segmentation contours is original and certainly appealing. Unfortunately the content of this work seems quite preliminary in terms of presentation and experiments.

First, the methodology is only sketched, while the motivation underlying the modelling rationale is often missing. For example, it is not clear what is the difference between "image evolution" and "contour evolution" models, besides the implementation details, and what motivates the definition of these two different modelling approaches in parallel.
Second, the experimental paradigm is controversial. The application on medical imaging is overly simplistic, as the authors do not consider the original 3D image stack, but rather the set of corresponding 2D slices modelled independently. The paper seems to ignore the large variety of body organ segmentation methods already available to the community, most of them working in 3D (e.g. [1-4]). The paper should necessarily compare with respect to these approaches and, even more importantly, with respect to standard level sets methods.

From the practical perspective, the proposed method builds upon the results obtained with the UNet, and therefore is characterised by an additional computational burden. Given that the the training of neural ODE is not straightforward and computational expensive, the use of this model for achieving a tiny accuracy improvement seems overkill for this kind of application. Moreover, the segmentation accuracy is still computed slice-by-slice in the 2D images, and no information is available for the consistency of the reconstruction in 3D (regularity over the vertical axis).

Finally, the results reported in Table 3 are not clear, why the metrics of the competing methods are approximated (e.g. ~ 0.7), while for the proposed methods are given up to the 3rd decimal term?


[1] 3D Kidney Segmentation from Abdominal Images Using Spatial-Appearance Models. Fahmi Khalifa, Ahmed Soliman, Adel Elmaghraby, Georgy Gimel'farb, and Ayman El-Baz 1.

[2] Automatic Detection and Segmentation of Kidneys in 3D CT Images Using Random Forests. Rémi Cuingnet, Raphael Prevost, David Lesage, Laurent D. Cohen, Benoit Mory, Roberto Ardon. Medical Image Computing and Computer-Assisted Intervention – MICCAI 2012: 15th International Conference, Nice, France, October 1-5, 2012, Proceedings, Part III

[3] Multi-organ localization with cascaded global-to-local regression and shape prior. Medical image analysis. Gauriau, R., Cuingnet, R., Lesage, D., & Bloch, I. (2015), 23(1), 70-83.

[4] Joint Classification-Regression Forests for Spatially Structured Multi-object Segmentation. Ben Glocker, Olivier Pauly, Ender Konukoglu, Antonio Criminisi. ECCV 2012 pp 870-881

**Experience Assessment:**

I have published in this field for several years.

**Review Assessment: Checking Correctness Of Derivations And Theory:**

I carefully checked the derivations and theory.

**Review Assessment: Checking Correctness Of Experiments:**

I carefully checked the experiments.

**Review Assessment: Thoroughness In Paper Reading:**

I read the paper thoroughly.

---

### Official Review · AnonReviewer1 · 2019-10-25
**Official Blind Review #1**

**Rating:** 3

**Review:**

This paper proposes to integrate Neural ODEs into image segmentation using level sets. I think the paper makes a good methodological contribution, however I don't think that ICLR is the best conference to publish this as the topic (image rather semantic segmentation) is too narrow and in my humble opinion, it won't attract the interest of ICLR audience. Moreover, it seems to me that the saliency object detection experiment is not a very convincing one as the methods compared are a bit old (mainly from 2015-2016).  I strongly recommend the authors try to publish this at MICCAI focusing on kidney segmentation or any other related medical imaging application.

**Experience Assessment:**

I have read many papers in this area.

**Review Assessment: Checking Correctness Of Derivations And Theory:**

N/A

**Review Assessment: Checking Correctness Of Experiments:**

I assessed the sensibility of the experiments.

**Review Assessment: Thoroughness In Paper Reading:**

I made a quick assessment of this paper.

---

### Public Comment · ~Yiping_Lu1 · 2019-09-27
**Related works**

Congrats on your work and I really enjoy reading it.
I'm writting the comment to introduce some of our related works on discretization Neural ODEs, control and computer vision applications
Lu Y, Zhong A, Li Q, et al. Beyond finite layer neural networks: Bridging deep architectures and numerical differential equations[J]. arXiv preprint arXiv:1710.10121, 2017.
Long Z, Lu Y, Ma X, et al. PDE-net: Learning PDEs from data[J]. arXiv preprint arXiv:1710.09668, 2017.
Zhang X, Lu Y, Liu J, et al. Dynamically unfolding recurrent restorer: A moving endpoint control method for image restoration[J]. arXiv preprint arXiv:1805.07709, 2018.


Also these early papers also aim to connect ODEs and deep learning
Weinan, E. "A proposal on machine learning via dynamical systems." Communications in Mathematics and Statistics 5.1 (2017): 1-11.
Li, Qianxiao, et al. "Maximum principle based algorithms for deep learning." The Journal of Machine Learning Research 18.1 (2017): 5998-6026.
Weinan, E., Jiequn Han, and Qianxiao Li. "A mean-field optimal control formulation of deep learning." Research in the Mathematical Sciences 6.1 (2019): 10.
Haber, Eldad, and Lars Ruthotto. "Stable architectures for deep neural networks." Inverse Problems 34.1 (2017): 014004.
Chang, Bo, et al. "Multi-level residual networks from dynamical systems view." arXiv preprint arXiv:1710.10348 (2017).
Ruthotto, Lars, and Eldad Haber. "Deep neural networks motivated by partial differential equations." arXiv preprint arXiv:1804.04272 (2018).

---

> ### Author Response · Authors · 2019-09-27
> **Related works: reply**
>
> Thank you for your kind words and for sharing your exciting work and additional references with us, Yiping Lu.
> We will update our manuscript's introduction to reflect this new information.

---

### Decision · Program_Chairs · 2019-12-19

**Decision:**

Reject

**Comment:**

This paper addresses the classic medial image segmentation by combining Neural Ordinary Differential Equations (NODEs) and the level set method. The proposed method is evaluated on kidney segmentation and salient object detection problems. Reviewer #1 provided a brief review concerning ICLR is not the appropriate venue for this work. Reviewer #2 praises the underlying concept being interesting, while pointing out that the presentation and experiments of this work is not ready for publication yet. Reviewer #3 raises concerns on whether the methods are presented properly. The authors did not provide responses to any concerns. Given these concerns and overall negative rating (two weak reject and one reject), the AC recommends reject.